
# Dynamic assessment of rainfall erosivity in Europe: evaluation of EURADCLIM ground-radar data

Francis Matthews[1], Pasquale Borrelli[1,2], Panos Panagos[3], Nejc Bezak[4]

[1] Department of Science, Roma Tre University, Rome, Italy
5 [2] Department of Environmental Sciences, Environmental Geosciences, University of Basel, Basel, Switzerland
[3] European Commission, Joint Research Centre (JRC), Ispra, Italy
[4] Faculty of Civil and Geodetic Engineering, University of Ljubljana, Ljubljana, 1000, Slovenia

*Correspondence to*: Nejc Bezak (nejc.bezak@fgg.uni-lj.si)

10  **Abstract.** Heavy rainfall is the main driver of water-induced soil erosion, necessitating accurate spatial and temporal predictions of rainfall erosivity to predict the soil erosion response. This study evaluates the ground radar-based EUropean RADar CLIMatology (EURADCLIM) precipitation grids to quantify rainfall erosivity across European countries. Compared to Global Rainfall Erosivity Database (GloREDa) gauge-based interpolations, EURADCLIM overpredicts rainfall erosivity, principally due to residual artefacts in some regions which inflate the instantaneous rainfall rates. Overprediction is most 15  pronounced in European regions with lower radar antenna coverage and complex topography, whereas flatter regions with lower erosivity and better radar coverage are better predicted. Disagreement attributes to the input radar quality in EURADCLIM (derived from OPERA) and to a lesser extent the uncertainty in GloREDa due to its limited gauge records in some regions. Event (EI30) time series analysis showed reasonably good performance (KGE > 0.4) in 50 % of the evaluated gauge locations, although significant overprediction by EURADCLIM was evident in the upper quantiles in some countries. 20  Accounting for the propagation of these remaining time-slice artefacts, which have a large impact on the temporally-aggregated R-factor, applying a 80 mm/h threshold to limit the maximum I30 value (i.e., less than 0.1% of GloREDa events exceed this threshold) during the calculation of rainfall erosivity significantly improves the performance of the EURADCLIM dataset at annual, monthly and event time scale. Following adjustment, EURADCLIM best agrees with GloREDa across Europe in July and August, while bigger differences were observed in June and winter in general. Annually, the spatially aggregated rainfall 25  erosivity per country had a percent bias below 10 %. While applying simple I30 thresholds is promising, radar artefacts remain significant in areas with lower quality rainfall retrievals. Notably, regions in Europe with lower quality or absent data furthermore coincide with established high soil erosion rates. In the absence of spatiotemporally continuous, high-quality ground-radar retrievals across Europe, we show the value of ensemble R-factor layers of EURADCLIM with three other rainfall erosivity grids (e.g., satellite retrievals) and discuss the possibility of ground radar to offer unique spatial detail in such 30  ensembles.



## 1 Introduction

Rainfall magnitude, duration, frequency and timing characteristics form the first order driver of soil erosion (Majhi et al., 2021). The extensively utilised rainfall erosivity index combines these characteristics into a statistical index representing the hydrometeorological forcings of rainfall and runoff, rendering it a critical data input for the Universal Soil Loss Equation (USLE) and its Revised (RUSLE) version (Renard et al., 1997). Independent of the chosen soil erosion model and the motivations for its application, accurate rainfall data inputs are an indispensable prior, particularly in model applications

predicting the multitemporal variability of soil erosion (Yin et al., 2017). The rainfall erosivity index (EIx: where x is typically 30 reflecting the maximum rainfall depth measured in 30 minutes) is characterized by high spatial and temporal variability (Bezak et al., 2021a, 2022; Fenta et al., 2023; Matthews et al., 2022; Panagos et al., 2022b), which is a product of the characteristics of rainstorm kinetic energy. At large spatial scales, high-frequency rain gauge data (i.e., ideally with 5-minute time step) with an adequate spatial density is needed to derive reliable long-term annual average rainfall erosivity (R-factor)

estimates (Fenta et al., 2023; Pidoto et al., 2022). For individual rainstorms, rain gauges are fundamentally represent point scale measurements with a limited cover limits predictions of the instantaneous spatial gradients of rainstorms in watersheds. Overcoming these scale limitations requires statistical interpolations based on process theory and/or remotely sensed proxy information, or stochastic rainfall generators. On top of these inescapable limitations on the quality of hydrometeorological forcings for erosion studies, the availability of suitable high-frequency rain gauge data is relatively low in many regions

(Panagos et al., 2017) and shows a globally decreasing trend (Sun et al., 2018). Several alternative approaches are available to estimate rainfall erosivity in data sparse regions, such as the erosivity density (ED) method to approximate the R-factor from the long-term annual average rainfall (Nearing et al., 2017; Panagos et al., 2016b; Yin et al., 2017), or remotely sensed precipitation datasets to estimate rainfall erosivity from high-temporal and often coarse-spatial resolution grids (Bezak et al., 2022; Chen et al., 2021; Delgado et al., 2022; Emberson, 2023; Fenta et al., 2023; Kim et al., 2020). In both cases, rain gauge

measurements are needed to derive reliable interpolations of ED or correct satellite-derived estimates of rainfall depth. Moreover, the information limitations within both approaches means that their accuracy can be expected to significantly decrease at finer temporal scales.

As climate change impacts precipitation characteristics around the globe (Hosseinzadehtalaei et al., 2020), rainfall erosivity patterns will change in the future (Panagos et al., 2022b). Changing magnitude, frequency and intensity characteristics in space

and time will interact with landscape disturbances such as cropping and tillage practices or forest fires to determine the spatial and temporal patterns of soil erosion. To properly capture the erosion response, rainfall erosivity maps need to be updated regularly with dynamic predictions of rainfall events. However, large-scale data collections (Panagos et al., 2017) are time-consuming when intermittent repetitions are required to collate offline data from national agencies. In recent years, satellite-based (Bezak et al., 2022; Emberson, 2023; Kim et al., 2020) and reanalysis-based (Matthews et al., 2022) estimates have



shown potential to move towards (near-)real time quantifications of the hydrological drivers of soil erosion. However, these alternative ways of producing rainfall erosivity maps yielded statistical disparities compared to gauge station quantifications (Emberson, 2023; Kim et al., 2020; Matthews et al., 2022), such as variability smoothing, missing events, and seasonal and/or spatial bias within the precipitation estimates. In the absence of high temporal resolution rain gauge data (ideally 5-minute), predicting the relationship between rainfall depth and rainfall erosivity presents a further challenge due to the high sensitivity

of the latter to the sub-timestep rainstorm intensity (Matthews et al., 2022). Therefore, more optimal approaches need to be tested for dynamic large-scale rainfall erosivity maps, which are reconcilable with catchment-scale simulations of soil erosion. In this respect, rainfall depth acquisitions from radar (RAdio Detection And Ranging) show promise due to their potential to resolve instantaneous rainstorm characteristics with high spatiotemporal detail. Within Europe, the European climatological high-resolution gauge-adjusted radar rainfall dataset (EUropean RADar CLIMatology (EURADCLIM)) (Overeem, 2022;

Overeem et al., 2023) may therefore show promise for producing rainfall erosivity predictions.

The primary aim of this study is to evaluate the performance of EURADCLIM ground-radar compilations to estimate the large-scale rainfall erosivity patterns in Europe at various timescales. Given the potential biases in EURADCLIM associated with artifacts in its 1-hourly time steps, this study further analysed the implications of imposing I30 threshold values to limit the influence of rainfall retrieval errors in EURADCLIM which can strongly influence the event-scale rainfall erosivity (EI30).

The R-factor derived from EURADCLIM was compared with global rainfall erosivity products (Bezak et al., 2022; Das et al., 2024) to evaluate the dis(agreement) in their pan-European R-factor patterns. Further insights are given into: i) the advantages and limitations of using EURADCLIM to estimate rainfall erosivity from the event to long-term annual average time step, and ii) the potential of multinational ground-based RADAR data with high spatial and temporal resolution to offer valuable information within ensemble rainfall erosivity predictions, based on IPCC-like principles, wherein differing precipitation

retrieval methods (e.g., satellite-based, ground radar-based, reanalysis) can be leveraged to indicate (dis)agreements in rainfall erosivity at large-scales.

## 2 Data and methods

### 2.1 GloREDa

To investigate the agreement between EURADCLIM and other gauge-based predictions, we used the GloREDa 1.2 average monthly and annual predictions provided by the European Soil Data Centre (Panagos et al., 2012, 2022a). We further used GloREDa 1.2's European gauge data, which is a subset of more than 3,939 stations from around the globe (Panagos et al., 2023), augmented with an additional 314 gauge stations compared to GloREDa 1.0 (Panagos et al., 2023), and including the Rainfall Erosivity Database at European Scale (REDES) for Europe (Panagos et al., 2015). GloREDa 1.2 therefore contains

information on over 300,000 erosive rainfall events (EI30) from more than 1,300 stations in Europe, calculated using the (R)USLE methodology (Ballabio et al., 2017; Panagos et al., 2015, 2023; Renard et al., 1997; Wischmeier and Smith, 1978).



The following information from GloREDa was used within this study: i) detailed time series information on specific erosive rainfall events such as date of the event, precipitation amount, kinetic energy, maximum 30-min rainfall intensity and event rainfall erosivity (Panagos et al., 2015) and ii) monthly and annual average rainfall erosivity maps that were produced from temporal aggregations of the erosive rainfall events included in GloREDa (Panagos et al., 2023). For time series comparisons, overlapping pan-European data year 2013 was used, augmented with Slovenian stations for period 2016-2020.

## 2.2 EURADCLIM

EUropean RADar CLIMatology (EURADCLIM) is a climatological dataset with ground radar rainfall accumulations at 1 hour and 24 hours and a spatial grid resolution of two kilometres (Overeem et al., 2023). Its second version was recently released, with temporal coverage 2013-2022, improved removal of non-meteorological echoes and better rain gauge coverage (EURADCLIM web-page, 2024). EURADCLIM is derived from the Operational Program on the Exchange of Weather Radar Information (OPERA) gridded composite radar dataset, which has a 15-minute temporal resolution and is sourced from 138 European radar antenna (Overeem et al., 2023). Ground radar offers highly valuable information for rainfall retrieval, however with numerous pitfalls, such as: i) general underestimations of rainfall of several percentage points, ii) overestimations during dry conditions due to artifacts (e.g., non-meteorological echoes), and iii) a range of other technical difficulties (radar beam attenuation, changes in the reflectivity profiles with distance from the antenna, calibration, among others) which incorporate noise within the retrievals. For secondary applications, EURADCLIM therefore implements numerous noise removal filters and processing steps on OPERA and combines it with data from 7,700 rain gauges included in the European Climate Assessment & Dataset (used in E-OBS) (Overeem et al., 2023).

For this study, hourly rainfall accumulations for the period 2016-2022 (i.e., prior 2016 radar coverage was smaller) were used to calculate the EI30 and to derive the aggregated monthly and annual rainfall erosivity maps. EURADCLIM version 2.0 was used since preliminary analysis indicated much smaller and more realistic rainfall erosivity values compared to version 1.0. It should be noted that 2013 was the only overlapping year between the GloREDa (for Europe) and EURADCLIM due to 75% of the GloREDa data being collected before 2000, which is one of the limitations of GloREDa (Panagos et al., 2023). Hence, rain gauges covering the year 2013 were used for the comparison at the event scale between GloREDa and EURADCLIM for multiple countries in Europe. Additionally, comparisons were performed over an extended time period for Slovenian stations which cover the period 2016-2020 in GloREDa.

## 2.3 Rainfall erosivity

Multiple computationally efficient options to calculate rainfall erosivity (EI30) through temporal disaggregation of hourly EURADCLIM were evaluated using time series data from the 62 Slovenian stations in GloREDa 1.2 (Panagos et al., 2023), covering the period 2010-2020. Slovenia has large spatial and temporal variability in rainfall erosivity, ranging values below





1,000 MJ mm ha$^{-1}$ h$^{-1}$ to more than 10,000 MJ mm ha$^{-1}$ h$^{-1}$ (Bezak et al., 2021b). Moreover, the countries climate spans Alpine,

Mediterranean and Temperate-continental zones (Dolšak et al., 2016). Given this climatological diversity, Slovenia was used as a case study to evaluate the most appropriate method to disaggregate hourly rainfall accumulation data to compute the event-scale rainfall erosivity (EI30).

Firstly, the EI30 was calculated from 30-min rainfall data (GloREDa 1.2 dataset) following the approach and equations described in Panagos et al., (2023). These EI30 values were considered as the ground-truth values for the 62 Slovenian stations.

Secondly, the measured 30-min rainfall data was aggregated to the hourly time step matching the EURADCLIM resolution, and four rule-based rainfall disaggregation schemes were tested: i) 50 % of rainfall occurs in first 30 minutes and 50 % in second 30 minutes; ii) 33.3 % of rainfall falls in first 30 minutes and 66.6 % in second 30 minutes; iii) 25 % of rainfall falls in first 30 minutes and 75 % in second 30 minutes; iv) 20 % of rainfall occurs in first 30 minutes and 80 % in second 30 minutes. EI30 was then recalculated using the 30-minute rainfall data from each scheme, giving percent biases of: i) -37 %; ii) -13 %;

iii) 1 %; iv) 10 %. Hence, scheme iii) was used to obtain 30-minute data (Figure S1). Additionally, we evaluated the performance of the conversion factors developed by (Panagos et al., 2016a) for computing EI30 based on the original hourly rainfall data. However, this approach overestimated the long-term rainfall erosivity by 40 % compared to the 30-minute rainfall data.

Following this test case, scheme iii) was applied (i.e., 25 % of rainfall falls in first 30-min and 75 % in second 30-min) to

compute the gridded EI30, and the limitations of transposing the method from gauge measurements to ground-RADAR based acquisitions are later discussed. Relevant considerations include the persistence of unfiltered artefacts in EURADCLIM, resulting in high estimates of the total kinetic energy (E) and maximum continuous 30-minute rainfall (I30). To investigate the influence of the latter, the predictive skill of EURADCLIM-based EI30 was evaluated based following the implementation of several I30 limits (20-300mm). The R code (R Core Team, 2021) for calculating rainfall erosivity developed by (Pidoto et al.,

2022) was used together with the equation from (Brown and Foster, 1987) for the calculation of the specific kinetic energy. All the equations used for the calculation of the rainfall erosivity are shown by Panagos et al., (2023).

## 3 Results and discussion

### 3.1 Annual average rainfall erosivity (R-factor)

Figure 1 compares the average annual rainfall erosivity, or the R-factor, based on the EURADCLIM and GloREDa datasets.

EURADCLIM overestimates rainfall erosivity in most of the comparable area in Europe (Figure 1 and Figure S2). Specifically, the R-factor for the GloREDa dataset (for the region shown in Figure 1) is around 719 MJ mm ha$^{-1}$ h$^{-1}$ with a standard deviation of around 537 MJ mm ha$^{-1}$ h$^{-1}$, while average annual rainfall erosivity for EURADCLIM is around 1,470 MJ mm ha$^{-1}$ h$^{-1}$ with a very high standard deviation of over 10,000 MJ mm ha$^{-1}$ h$^{-1}$. In both cases, the analysis was limited to countries with EURADCLIM coverage, omitting countries such as Italy, Greece and Lithuania. For most of the European countries,



EURADCLIM yields higher or even much higher R-factor values, especially in Croatia, Bosnia and Herzegovina, Serbia, and Estonia (Figure S2). The Pearson correlation coefficient between EURADCLIM and GloREDa country-averaged R-factor predictions is modest (r = 0.24), with a percent bias of 96 % (Table 1), indicating significant overestimation by EURADCLIM and exaggerated spatial variability (Figure S2). Excluding countries with large R-factor disparities (i.e., Croatia, Bosnia and Herzegovina, Serbia, and Estonia) reduced percent bias reduced to 50 % (Figure 1 and Figure S2). THis error principally

attributes to the heterogenous radar antenna coverage in OPERA (Saltikoff et al., 2019) and the sparsity of the rain gauge network included in EURADCLIM in Southern and Eastern Europe (Overeem et al., 2023). An additional factor in such regions is high topographical complexity, which would ideally require high radar antenna coverage and accompanying rain gauge measurements for reliable rainfall acquisitions. Within this comparison it should be noted that some countries had a highly suboptimal rain gauge coverage in GloREDa (Croatia and Estonia) or were completely interpolated (i.e., Bosnia and

Herzegovina, Serbia) due to an absence of gauging stations (Overeem et al., 2023; Panagos et al., 2023).

As discussed in section 3.3, large differences between the EURADCLIM and GloREDa R-factors attribute to a relatively small number of overpredicted extreme EI30 values in EURADCLIM in some regions. Indeed, applying a limit of 80 mm/h to the I30 parameter brought the prediction skill of the R-factor in Europe in line with other predictions such as GloRESatE, IMERG and COMRPH (Table 1 and section 3.4). With respect to GloREDa, the maximum 30-minute rainfall is higher than the 80

mm/h threshold only for a relatively small number of events (i.e., less than 0.1%). Although Overeem et al., (2023) indicated that EURADCLIM can capture extreme precipitation events, strong regionality remains in the predictions (Overeem et al., 2023). For countries such as Finland, Norway or Slovenia, the R-factor is better predicted (Figure 1, Figure S2), which corresponds to a better agreement in the total precipitation by the EURADCLIM and E-OBS datasets (Overeem et al., 2023). An additional spatial comparison for Austria and Poland is provided (Figure S3 and Figure S4, respectively) which show

relatively good agreement in the spatially-aggregated average annual rainfall erosivity (i.e., Austria: GloREDa R: 1,170 MJ mm ha$^{-1}$ h$^{-1}$ and EURADCLIM R: 1,320 MJ mm ha$^{-1}$ h$^{-1}$; Poland: GloREDa R: 554 MJ mm ha$^{-1}$ h$^{-1}$ and EURADCLIM R: 744 MJ mm ha$^{-1}$ h$^{-1}$), but poor spatial distributions due to remaining artefacts, giving unrealistic spatial patterns compared to predictions from GloREDa and CMORPH (Figure S5). While the spatial patterns are slightly better preserved for Poland (Figure S4), in case of Austria (Figure S3) there is a clear impact of unfiltered radar echos. Similar, issues can also be detected

in some other countries like Spain, Romania, and other areas of South-Eastern Europe (Figure 1), which are likely caused by a high artefact presence due to beam blockage or other errors (Overeem et al., 2023).



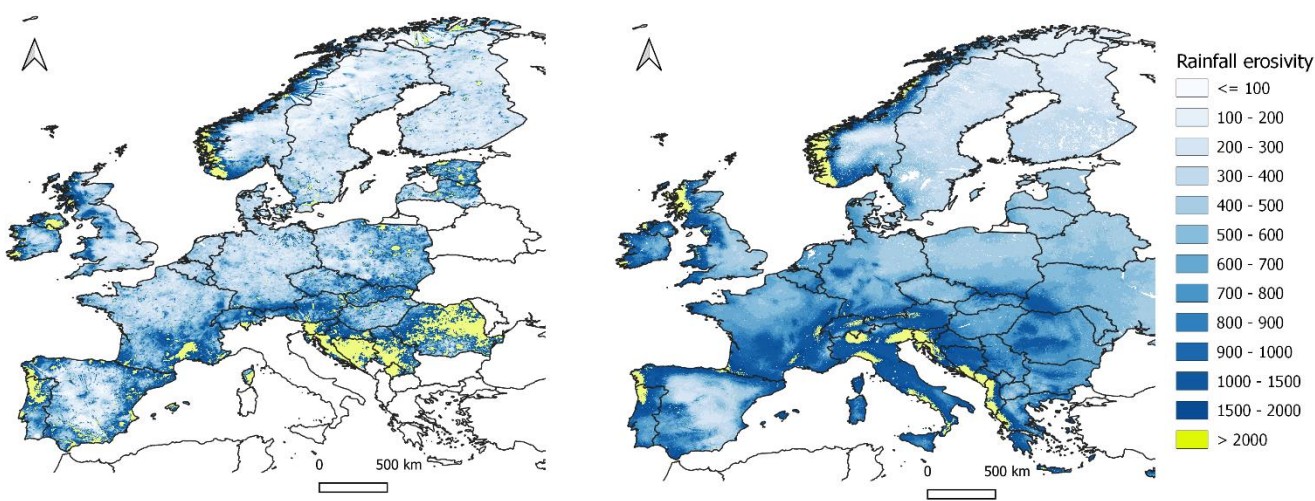

**Figure 1: Comparison between annual rainfall erosivity (R) (MJ mm ha⁻¹ h⁻¹) calculated using EURADCLIM dataset (left) and GloREDa dataset (right) for Europe.**

**Table 1: Coefficient of determination (R2), percent bias (PBIAS) and Mean Error (ME) values for annual rainfall erosivity for spatially aggregated countries (country-average values were used) covered by EURADCLIM. Comparison between GloREDa (Panagos et al., 2023), GloREDatE (Das et al., 2024), IMERG (Das et al., 2024) and CMORPH (Bezak et al., 2022) is shown. Additionally, the EURADCLIM performance using the I30 threshold is shown (Section 3.4).**

| GloREDA | EURADCLIM | GloRESatE | IMERG | COMRPH | EURADCLIM I30-threshold |
|---|---|---|---|---|---|
| R2 | 0.06 | 0.71 | 0.62 | 0.67 | 0.66 |
| PBIAS (%) | 96 | -25 | 10 | -8 | 9 |
| ME [(MJ mm yr ha⁻¹ h⁻¹)] | 890 | -241 | 96 | -79 | 83 |

A multi-platform comparison was also made between the GloREDa (Panagos et al., 2023) and GloRESatE (Das et al., 2024), IMERG (Das et al., 2024) and CMORPH (Bezak et al., 2022) datasets (Table 1, Figure S2, Figure S5). The agreement between CMORPH and IMERG and GloREDa was better compared to the EURADCLIM. Interestingly, slightly worse performance was observed for GloRESatE, measured by the percent bias and mean error, compared to CMORPH and IMERG. This is despite GloRESatE being based on the CMORPH, IMERG and ERA5-Land (Das et al., 2024). Thus, it seems that satellite-based products like CMORPH (Bezak et al., 2022) or IMERG (Emberson, 2023) should be preferred compared to radar-based product like EURADCLIM for applications focussing on pan-European coverage. However, it should be noted that Europe was the continent where the best agreement was found between the CMORPH and GloREDa (Bezak et al., 2022). Hence, different results could be obtained in other regions.



## 3.2 Monthly rainfall erosivity

The monthly rainfall erosivity magnitudes derived from EURADCLIM follow the seasonal trends in GloREDa (Panagos et al., 2023), wherein the average the summer (June-July-August) rainfall erosivity is around 3-4 higher than winter (December-January-February). However, significant positive seasonal bias in the monthly and seasonal averages was present in the original EURDACLIM rainfall erosivity predictions (Table. 2). For example, EURADCLIM produced a summer average of 800 MJ mm ha$^{-1}$ h$^{-1}$, which is approximately 2.5 times higher than the GloREDa prediction (Panagos et al., 2023). The average winter value (180 MJ mm ha$^{-1}$ h$^{-1}$) is similarly inflated, at roughly double the GloREDa prediction. As in the case of the annual R-factor, seasonal overestimation is more pronounced in areas of Europe with a generally higher rainfall intermittency and erosivity (i.e. Southern Europe), compared to lower erosivity areas in Northern Europe (i.e., less than 100 MJ mm ha$^{-1}$ h$^{-1}$ month$^{-1}$) (Figure S6).

Compared to the unadjusted EUDCLIM simulations, CMORPH for example (Bezak et al., 2022), yields a better monthly agreement with GloREDa with a coefficient of determination ranging from 0.68 to 0.95 and percent bias from -47 % to 110 % (mean = 23 %). However, as in the case of the annual R-factor (section 3.1), a significant improvement in the monthly coefficient of determination ($R^2$ = 0.49 to 0.94) and % bias (-15 % to 103 %) could be achieved when applying a limit (80 mm/h) to the I30 parameter when calculating EI30 from EURADCLIM (section 3.4; Table 2). Thus, further filtering of extreme outliers in EURADCLIM shows the potential to bring the monthly predictive skill of EURADCLIM in line with satellite-based retrievals, however both seasonal and spatial disparities in performance require consideration (Figure 1 and Table 2).

**Table 2: Coefficient of determination (R2) and percent bias (PBIAS) values for the monthly rainfall erosivity values between EURADCLIM and GloREDa for countries (country-average values were used) covered by both datasets. Additionally, the EURADCLIM performance using the I30 threshold is shown (Section 3.4).**

| EURADCLIM | Jan | Feb | Mar | Apr | May | Jun | Jul | Aug | Sep | Oct | Nov | Dec |
|---|---|---|---|---|---|---|---|---|---|---|---|---|
| R2 | 0.85 | 0.66 | 0.41 | 0.60 | 0.67 | 0.17 | 0.40 | 0.16 | 0.55 | 0.40 | 0.60 | 0.46 |
| PBIAS (%) | 61 | 227 | 53 | 18 | 3 | 267 | 17 | 61 | 17 | 212 | 93 | 225 |
| EURADCLIM I30-threshold | Jan | Feb | Mar | Apr | May | Jun | Jul | Aug | Sep | Oct | Nov | Dec |
| R2 | 0.83 | 0.86 | 0.74 | 0.94 | 0.76 | 0.49 | 0.58 | 0.68 | 0.73 | 0.61 | 0.64 | 0.67 |
| PBIAS (%) | 46 | 103 | 25 | -13 | -15 | 14 | -10 | -2 | -9 | 14 | 48 | 86 |

## 3.3 Event rainfall erosivity

EURADCLIM-derived EI30 values were compared with GloREDa measurements for all comparable (i.e. temporal matches within a 24-hour window) simulations and gauge measurement locations in 2013 (Figure 2). For the 6,262 events, Figure 2 firstly shows the existence of potentially large positive discrepancies (> 1000 MJ mm ha$^{-1}$ h$^{-1}$) between EURADCLIM and GloREDa, occurring in a small minority of events. These overpredictions likely attribute to unfiltered artefacts (false positives),



which result in high error in multiple sites concurrently, particularly in Spain (ES) and Romania (RO) which cause large
discrepancies in their upper quantiles (Figure S7). Conversely, negative errors showed less temporal correspondence between
gauges, indicative of localised underpredictions which may relate to missed events (false negatives) or geolocation issues in
EURADCLIM when resolving the spatial rainfall intensity gradients. In this respect, differences in the spatiotemporal
continuity of the OPERA radar network (Saltikoff et al., 2019) and the clutter-removal algorithm applied to EURADCLIM
(Overeem et al., 2023) may be a source of these underestimations, if artefacts were falsely classified and removed. The overall
effect of these complex errors, amplified by the sensitivity EI30 to overestimations at singlular time steps, creates a bias
favouring overprediction especially in summer months (Section 3.2). Further analysis of the relative error allowed preliminary
baseline quantifications based on the sample of events, showing that 50 % of EURADCLIM derivations of EI30 have a relative
error of 35 %, 75 % with an error below 59 %, and 95 % with an error below 88 %. Below a 100 % error, there is little
systematic tendency for under or over prediction, therefore suggesting that artefacts in the EI30 values influence the upper-
most quantiles via a small but critical addition of high magnitude events. Despite limitations, EURADCLIM produced
reasonably good predictive performances (Kling–Gupta efficiency (KGE) > 0.4) for 50 % of locations with over 10 comparable
EI30 events (n = 231), with all sites producing a KGE > -0.41 (equivalent to a Nash-Sutcliffe efficiency of 0).







**Figure 2: Event based comparison of rainfall erosivity (EI30) for GloREDa and EURADCLIM datasets for year 2013: a) the event-scale error in the EI30 prediction for positive and negative absolute errors (points) and their monthly average profiles (lines) with standard deviation (envelopes), b) the probability distribution of relative % error for positive (overpredictions) and negative (underpredictions) error, c) the cumulative % of GloREDa locations with a given Kling–Gupta efficiency (KGE), and d) the average KGE per country based on the average of the evaluated locations with GloREDa data.**

The event-scale analyses provided aim to give the most objective possible overview of the capability of EURADCLIM for EI30. However, grid-to-point comparisons exhibit fundamental differences due to the simplified representations of spatial and temporal scales in the former (Tozer et al., 2012). The overwhelming benefit of the continuous spatiotemporal acquisitions made by EURADCLIM is their capacity to resolve the spatial detail of storm cells determining the erosion response. For





instance, Figure S8 shows the spatial patterns of EI30 for an event on 20ᵗʰ of June 2013 in Germany for which the agreement between EURADCLIM and GloREDa is relatively good despite underestimation at a few gauge sites. However, preceding this event were relatively extreme floods occurring at the end of May and early June 2013 in Germany (Thieken et al., 2016). In this case, all but one gauge in GloREDa recorded a value below 200 MJ mm ha$^{-1}$ h$^{-1}$ over multiple days, which represents a smaller event within GloREDa compared to 20ᵗʰ of June, although the latter had no associated flooding. This event represents

an example in which relatively low GloREDa coverage in Central Germany meant that the main peak of the storm contributed proportionately much less to the long-term rainfall erosivity and all the entire event was split in several smaller erosive events. Considering the impacts of these spatial mismatches over short time periods, datasets with remotely acquired rainfall such as EURADCLIM are critical to acquire representative predictions of soil erosion at large-scales.

        Additionally, visualisation of the EI30 spatial gradients combined with Sentinel-2 data on the soil cover condition offers

insights into the use of EURADCLIM for instantaneous erosion mapping. Figure 3 (upper) shows two significant erosive rainfall events that occurred in UK in February 2020 (Sefton et al., 2021) and in France in October 2020 (Storm Alex, 2024). EURADCLIM can detect spatial gradients in rainfall erosivity down to a fine (multi-kilometre) resolution which can greatly benefit process-based to empirical erosion model applications, as well as machine learning (data-driven) algorithms for (spatiotemporal) erosion feature detection (Shmilovitz et al., 2023). The combination of spatial EI30 predictions with Sentinel-

2 NDVI data shows the possibility to identify spatiotemporal coincidence between high rainfall intensity and arable fields at bare or low crop development stages. For example, in South-West England, the coincidence between erosive rainfall centres (i.e., > 100 MJ mm ha$^{-1}$ h$^{-1}$) and at-risk arable land was relatively low (Figure 3). In contrast, the example in South-West France shows spatial coincidence between heavy rainfall (i.e., > 300 MJ mm ha$^{-1}$ h$^{-1}$) and clusters of fields with particularly low vegetation cover, principally due to relatively recent seed bed preparation of winter crops around October (Figure 3). Field

evidence (Boardman and Favis-Mortlock, 2014) highlights the time-dependency of erosion, wherein spatiotemporal correspondences between tilled soil and heavy rainfall generate substantial soil loss. EURADCLIM may excel in detecting spatial detail in small-scale extreme events where a sub optimally distributed ground-based precipitation measuring network would otherwise be insufficient.



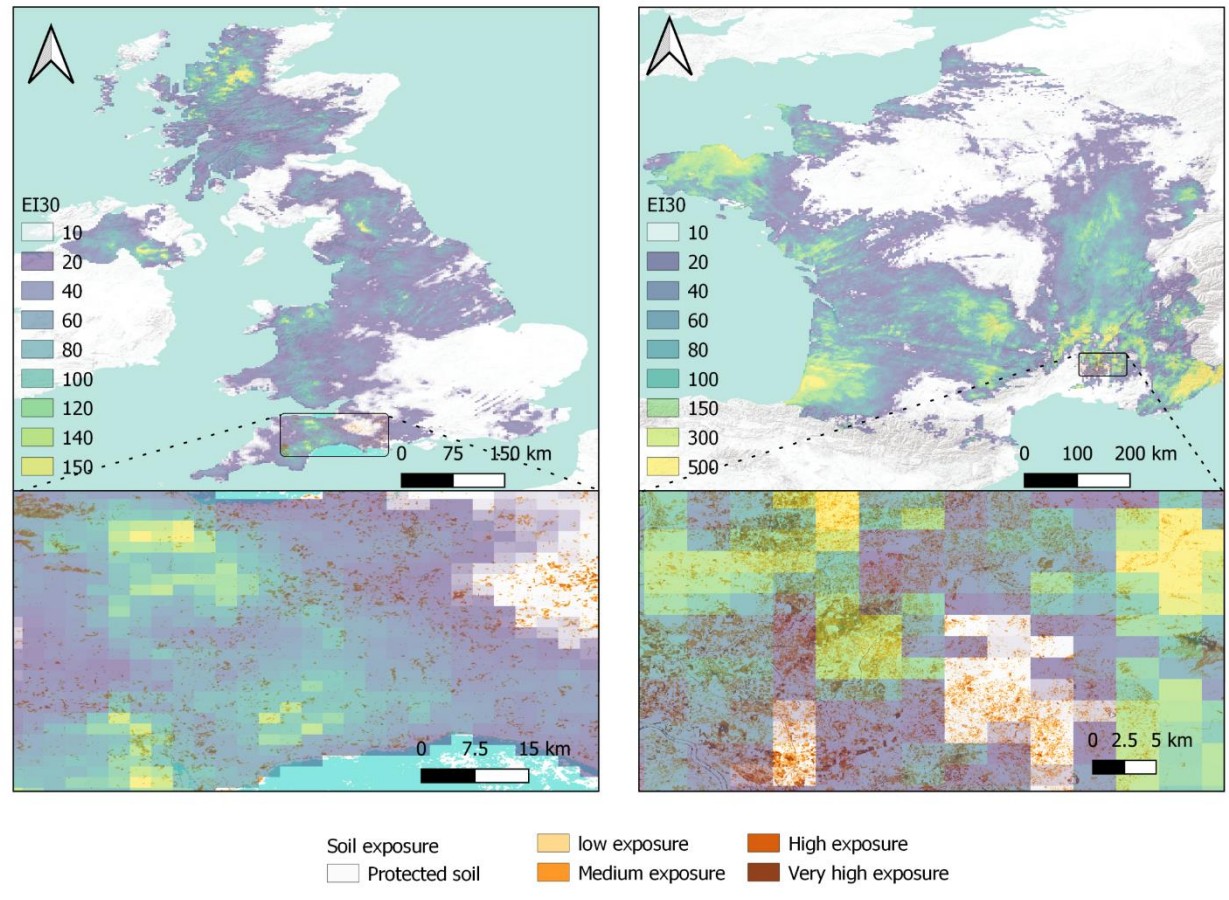

**Figure 3: Event based comparison of EURADCLIM dataset and soil exposure data derived from Sentinel-2 for two specific extreme events that occurred in UK in February 2020 and in France in October 2020. The relative exposure is approximated using categories of: protected soil (NDVI = 0.8 - 1), low exposure (NDVI = 0.6 - 0.8), medium exposure (NDVI = 0.4 - 0.6), high exposure (NDVI = 0.2 - 0.4) and very high exposure (NDVI = 0 – 0.2) based on the closest available Sentinel-2 acquisition to the event date.**


## 3.4 EURADCLIM bias correction

Individual outliers from radar-related artefacts and their interaction with temporal disaggregation methods strongly impact the agreement of the EURADCLIM dataset with the GloREDa (i.e., Figure S9, Figure S10, Figure S11). Consequently, at the event scale, a large inflation in the absolute % error on EI30 occurs for events in which the initial error on the precipitation depth is high (Figure S9). To improve the pan-European R-factor, which is a long-term statistical aggregation of individual EI30 events (Section 3.3), we additionally analysed the potential of I30 thresholds to limit large predictive errors by mitigating against artificially extreme EI30 values. Data for year 2013 for multiple GloREDa stations and data from Slovenian stations for the 2016-2020 was used to test the impact of different I30 thresholds on the KGE (Figure S12).



Based on the evaluation it was found that the I30=80 mm/h threshold, in which all I30 values exceeding this threshold were limited to 80 mm/h, can significantly improve the agreement between the EURADCLIM and GloREDa (Figure S12). We

argue that this threshold removes little to no actual extreme rainfall events due to the incredible rarity of an hourly rainfall rate exceeding 80 mm/h (Bezak et al., 2020, 2023; Mohr et al., 2020; Reder et al., 2022; Rusjan et al., 2009). Following limitation of the I30, a significant improvement was found in both the annual and monthly predictions when evaluated against GloREDa (Tables 1 and 2), which meant a superior performance compared to GloRESatE and IMERG datasets and a comparable performance to CMORPH (Table 1, Table 2, Figure S10). The mean annual rainfall erosivity (728 MJ mm ha$^{-1}$ h$^{-1}$) converged

on that of GloREDa, although still with a higher standard deviation of 945 MJ mm ha$^{-1}$ h$^{-1}$ (Figure 4). Moreover, both the average annual and monthly correspondence between EURADCLIM and GloREDa significantly improved across the whole domain (Table 2) and across countries (Figures 4 and 5), particularly for warmer part of the year compared to the colder season (Figure 5). Despite improvement, the EURADCLIM R-factor still has visibly remaining artefacts and is still overpredicted for countries like Bosnia and Herzegovina, Croatia, Serbia and Romania. Stricter limits on I30 (e.g. 20 mm/h) can filter be used

to filter a larger number of potential artefacts and provide smoother R-factor surfaces (Figure 4), however with the risk of impacting true high-intensity events, for which ground-radar is arguably most beneficial. As discussed in section 3.1 part of this overprediction can be related to the variable input data quality within EURADCLIM (Randeu and Schonhuber, 2000), which may limit the absolute potential of post subsequent corrections in some European regions.






**Figure 4: Above: The EURADCLIM annual rainfall erosivity (MJ mm ha⁻¹ h⁻¹) map in case of applying the I30 threshold value of 80 mm/h (a) and 20mm/h (b) to the EI30 to limit the influence of possible outliers (artefacts) from the radar data. Below: The absolute difference in the R-factor compared to the original value is given when applying a I30 threshold value of 80 mm/h (c) and 20mm/h (d) to the EI30.**






Figure 5: Comparison between the corrected EURADCLIM dataset (i.e., using the I30 = 80 mm/h threshold) at annual (upper panel) and monthly time step lower panel). Only European countries covered by EURADCLIM are shown (one point-one country).





## 3.5 Study limitations and the potential of ground radar in rainfall erosivity ensembles

Highlighting several methodological limitations is relevant for advancing pan-European ground-RADAR applications. Firstly, the 1-hourly EURADCLIM dataset required temporal disaggregation to calculate the EI30 parameter (section 2.3), but with
several possible limitations: i) the simple disaggregation scheme (i.e., 25 % of rainfall was considered in first 30-min and 75 % of rainfall in second 30-min) from hourly into a 30-minute resolution was satisfactory for the tested subset in Slovenia (Figure S1), however its performance in other climatic regions of Europe is not known, and ii) artefacts within EURADCLIM can be exaggerated by the disaggregation method which can create extreme artificial rainfall intensity peaks influencing the EI30 values (Figure S9). Addressing the former point (i) requires more high-resolution time series data from other stations
included in the GloREDa to investigate spatial variations in the potential error. The latter point (ii) is complex and relates strongly to the processing steps to remove non-meteorological echoes within EURADCLIM. Stricter manipulation of the EI30 equation, such as an I30 limit below 80 mm/h (Figure 4), may reduce the propagation of non-meteorological noise into the R-factor, but impact the predicted magnitudes of true event (type II error). Furthermore, related to both points is the selected EI30 equation (i.e., Brown and Foster, (1987)) which can further impact the R-factor due to its sensitivity to rainfall intensity
peaks (McGehee et al., 2021).

Secondly, despite GloREDa's use as a baseline estimate for comparison, it has potential recognised uncertainties (Ballabio et al., 2017; Bezak et al., 2022; Panagos et al., 2015, 2023), such as the mismatches in data periods used between stations and the differing gauge measurement resolutions between stations. Further mismatch is introduced due to GloREDa and EURADCLIM covering predominantly different periods (i.e., most of the GloREDa in Europe: 1951-2013; EURADCLIM:
2013-2022, in this study 2016-2022 period was used), which limits the number of comparable EI30 events. Mismatches in data periods may further induce issues of non-stationarity into comparisons of the long-term R-factor due to climate change. However, with a relatively large (n = 6,262) combined sample of EI30 events over which to make a direct comparison was possible (Section 3.3), key insights into the spatial difference in prediction capacity and the effects of radar artifacts were possible. Among the varying relevant considerations, the impact of precipitation artefacts recognised and addressed to some
extent in EURADCLIM (Overeem et al., 2023, 2024) remains a critical limitation for applications relying on rainfall intensity approximations such as rainfall erosivity.

In recent years different meteorological datasets have been used to derive rainfall erosivity, each with its unique advantages, limitations and uncertainties. Baseline interpolated predictions such as GloREDa have their own set of limitations given the complex spatial and temporal dynamics of rainfall. Hence, rainfall erosivity estimations could benefit from statistical
ensembles which capitalise on the agreement and disparities between different prediction methods. To give preliminary insights into multi-platform ensembles, we used GloREDa (Panagos et al., 2023), CMORPH (Bezak et al., 2022), IMERG (Das et al., 2024), GloRESatE (Das et al., 2024) and EURADCLIM (this study) to create a multi-product ensemble at a common resolution (i.e., 0.1° used by GloRESatE). The spatial patterns of the ensemble are shown in Figure 6 without (left) and with (right) EURADCLIM. The median ensemble without EURADCLIM shows generally smooth patterns in the spatial





dis(agreement) as quantified by the standard deviation (Figure 6), with most variability in areas of Southern Europe and the
Atlantic coast. Conversely, the addition of EURADCLIM into the ensemble adds significantly more spatial detail to the
patterns of disagreement between datasets (Figure 6). A visible component of this disagreement attributes to remaining
artefacts (e.g., linear features), however the addition of real features such as fine scale convective precipitation cells is a
potentially large benefit of ground radar data. Future studies are needed to obtain more comprehensive ensembles of rainfall

erosivity which include a wider variety of precipitation retrieval methods and EI30 calculation routines, as well as at varying
timescales (e.g., event/multi-day, monthly and annual scale) to match the hydrometeorological forcing requirements of erosion
models. However, fundamental consideration in such ensembles should be the assimilated rain gauge data within each gridded
dataset, the consideration of optimal heterogeneity between inputs (e.g. GloRESatE is based on both IMERG and CMORPH
data), and the necessity to incorporate fine spatial detail into such ensembles (e.g. through ground radar).


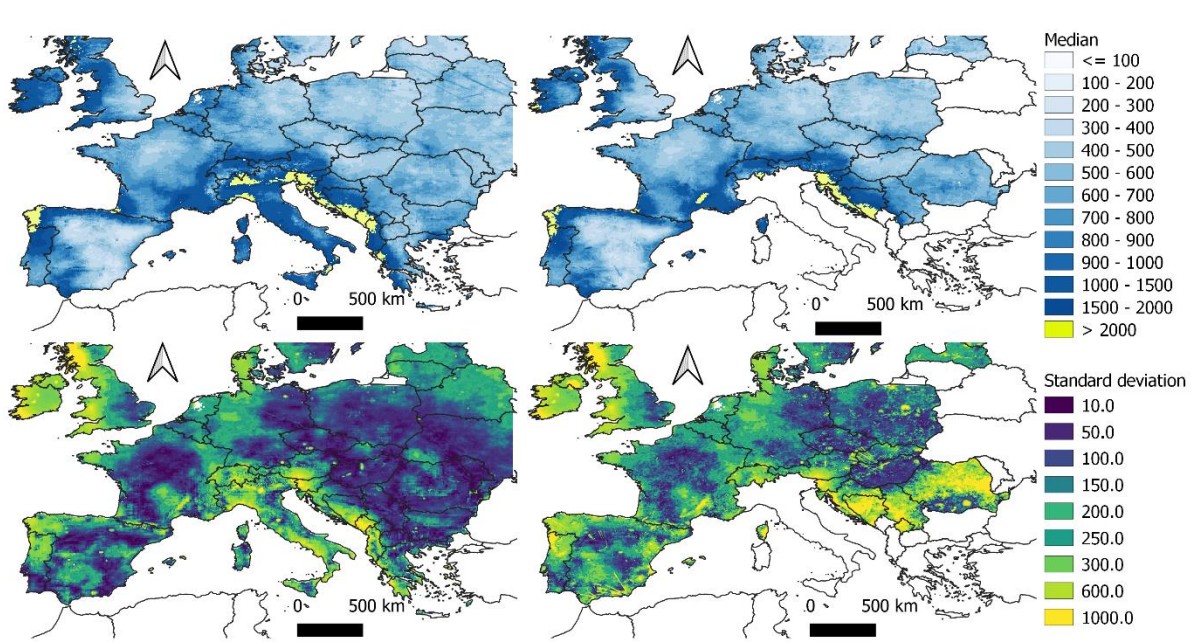

**Figure 6: Median and standard deviation (MJ mm ha⁻¹ h⁻¹) of the annual rainfall erosivity calculated based on the ensemble of different rainfall erosivity products: left) a 3 data source ensemble of GloREDa (Panagos et al., 2023), CMORPH (Bezak et al., 2022), IMERG (Das et al., 2024), GloRESatE (Das et al., 2024), right: a 4 data source ensemble of GloREDa (Panagos et al., 2023),**
**CMORPH (Bezak et al., 2022), IMERG (Das et al., 2024), GloRESatE (Das et al., 2024) and EURADCLIM (this study)).**



## 4 Conclusions

Based on the evaluation of EURADCLIM 1-hourly rainfall acquisitions to derive rainfall erosivity across multiple temporal scales, the following conclusions are drawn:

i)   EURADCLIM overestimates rainfall erosivity compared to GloREDa, principally due to the propagation of
artificially high rainfall rate predictions into the EI30 parameter. This overestimation was significant in regions like the Balkans, with complex topography and a low radar antenna coverage within the OPERA radar networks which limits spatially continuous application of EURADCLIM in Europe. Consequently, satellite-based products such as CMORPH with 30-minute acquisitions are more suitable for spatially continuous, large-scale rainfall erosivity estimations.

ii)   Despite the strong influence of non-meteorological artefacts on rainfall erosivity, EURADCLIM offers unique spatial detail to detect small-scale rainfall features (e.g., convective cells) critical for prediction erosion in susceptible fields. Future removal of non-meteorological echoes in EURADCLIM updates and a better quantification of its spatial error will augment its practical application in large-scale soil erosion prediction applications.

iii)   Given the strong impact of residual radar artefacts in EURADCLIM on EI30, rainfall erosivity (statistical sums of EI30 over time) estimates should account artificially high instantaneous rainfall rate predictions in the computation of EI30. Applying the simple threshold value of 80 mm/h to limit unrealistic I30 values significantly improves the performance of EURADCLIM dataset compared to the GloREDa. Stricter, spatially variable limits, or other methods of spatial smoothing for the R-factor, may further improve the quality of final map products.

iv)   Based on the different rainfall erosivity products, an ensemble (median and standard deviation) was derived to give first insights into a potential future avenue for updatable pan-European rainfall erosivity predictions. Ensembles will better allow the incorporation of uncertainty in the R-factor due to differing precipitation retrieval methods and the computation of EI30. As an ensemble component, EURADCLIM may offer unique spatial detail on rainfall rates which is unobtainable from other retrieval methods but critical for soil erosion prediction.


**Author contributions**

NB and FM developed the concepts of the manuscript, NB conducted most of the calculations and wrote the first draft. FM conducted the event-scale comparison. FM edited and improved the manuscript and figures. PB and PP edited and improved the manuscript and provided access to GloREDa dataset.


**Competing interests**

Authors declare no competing interests.





**Acknowledgements**

We thank the Royal Netherlands Meteorological Institute (KNMI) for providing the EURADCLIM dataset and ESDAC for making GloREDa dataset publicly available.

**Financial support**

Nejc Bezak is grateful for the support of the Slovenian Research Agency (ARRS) through grants no. P2-0180, J6-4628, J2-
4489, N2-0313 and the support from the UNESCO Chair on Water-related Disaster Risk Reduction. FM was co-funded by the Horizon Europe projects Soil O-LIVE (Grant No. 101091255) and AI4SoilHealth (Grant No. 101086179). PB received funding from the Swiss State Secretariat for Education, Research and Innovation (SERI), grant agreement no. 101086179, AI4SoilHealth.

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
