# Peer review of "Dynamic assessment of rainfall erosivity in Europe: evaluation of EURADCLIM ground-radar data"

_Hydrology and Earth System Sciences, 2024_

## Author Response (AR1)

Ljubljana, 24th of July 2025

TO:

Editorial Office Hydrology and Earth System Sciences

Dear Editorial Office,

Please find enclosed the revised version of the original manuscript entitled "*Dynamic assessment of rainfall erosivity in Europe: evaluation of EURADCLIM ground-radar data*" authored by F. Matthews, P. Borrelli, P. Panagos, and N. Bezak.

We would like to thank the editor and the reviewers for their time and efforts in reviewing our manuscript, for the constructive comments and suggestions that can help us bring the manuscript up to the Hydrology and Earth System Sciences journal standard, and for the opportunity to address the highlighted concerns and suggestions in this resubmission. We also would like to thank the editor Dr. Nadav Peleg for giving us chance to revise our manuscript and for his positive evaluation.

In response to the reviewers' comments, we have conducted a major revision and incorporated new material in the revised version. Most notably: (i) new analyses were conducted and included in the manuscript; (ii) additional information about specific research steps (e.g., disaggregation, rainfall erosivity calculation, GloREDa database) was added to the manuscript; (iii) several figures were modified; and (iv) several minor corrections that were suggested by two reviewers and Shuiqing Yin were addressed.

Please find below a detailed response to the reviewer's comments. All changes are indicated with red colored text in the manuscript.

Sincere regards on behalf of all the authors,

Nejc Bezak

**Shuiging Yin, 05 Feb 2025**

SYC1: Radar-based datasets offer the advantage of high temporal and spatial resolution, making them crucial for capturing the high variability of rainfall erosivity, particularly at the event scale. This study aims to evaluate the ground radar-based EURADCLIM precipitation dataset, which has an hourly temporal resolution and a 2 km spatial resolution, to quantify rainfall erosivity at annual, monthly, and event scales across European countries—a significant endeavor.

SYC1 Response: Thank you on behalf of all coauthors for the positive comments on the research endeavor.

SYC2: However, several concerns need to be addressed:

EURADCLIM is a ground radar-based dataset that has been bias-corrected using station observations. However, there is no clear information on whether the stations used in the development of EURADCLIM are the same as those in GloREDa. If they are, a separate discussion is needed, as data accuracy is generally higher in areas with stations and lower in regions without them.

SYC2 Response: Thanks for this relevant comment. In the revised version we enhanced the discussion regarding the overlap of EURADCLIM and GloREDa in terms of stations (and period) in section 2.2. A separate paragraph was added to discuss these important aspects.

SYC3: GloREDa's data is interpolated from station observations. Due to the limited density of stations, it provides a reasonable representation of long-term average rainfall erosivity. However, for event-scale and daily-scale rainfall erosivity, GloREDa is likely to have significant errors in areas without stations due to the high spatial variability of precipitation, particularly from heavy convective storms, at these finer scales. Therefore, using GloREDa as the ground truth to assess the accuracy of EURADCLIM should be approached with great caution.

SYC3 Response: Thank you for the comment. Perhaps some misconceptions are present that we clarified here and within the revised manuscript. When evaluating EURADCLIM predictions of rainfall erosivity, long-term spatially continuous comparisons were made by comparing the R-factor (i.e., the long-term annual average rainfall erosivity). All other comparisons were done based on grid-to-point comparisons between EURADCLIM pixels and GloREDa gauge station measurements, which allows multi-temporal comparisons of the EI30 and multiple levels of temporal aggregation. This is also more clearly explained in the revised version of the manuscript to avoid confusion (Sections 2.1 and 2.2).

SYC4: Additionally, some studies have used radar-based datasets to quantify rainfall erosivity, but they do not appear to be included in the review section. For example, the daily erosion model hosted by Iowa State University could be relevant to this study. SYC4 Response: Thank you. We included three additional references related to the rainfall erosivity estimation using radar-based approaches in the revised version of the manuscript.

SYC5: Other suggested improvements include:

- Adding the definition of the Kling-Gupta Efficiency (KGE).
- Including a sample event illustrating the dynamic evolution of rainfall erosivity.

SYC5 Response: Thanks for the suggestions. In the revised version we did:

- Define the Kling-Gupta Efficiency (KGE) acronym in abstract.
- Include an example of a specific event(s) showing the rainfall erosivity temporal distribution based on the EURADCLIM and GloREDa. Hence, a new figure was added to the Supplementary material and additional discussion was included in the manuscript.

**Anonymous Referee #1, 06 Feb 2025**

R1C1: I reviewed the manuscript "Dynamic assessment of rainfall erosivity in Europe: evaluation of EURADCLIM ground-radar data", and read it with interest, revealing a new potential use case for European gauge-adjusted radar data. The manuscript is well written and concise and considers the use of a new European gauge-adjusted radar precipitation dataset for erosivity modelling. I do have a couple of questions and comments that will hopefully help to authors to improve their manuscript, especially regarding the results and discussion.

R1C1 Response: We would like to thank Reviewer #1 for his/her positive evaluation of our study and useful suggestions provided. Please find below the response to specific comments.

**Introduction:**

R1C2: - L. 40-43: Not being an expert in erosivity: the rainfall erosivity index is, through high spatial and temporal variability a product of the characteristics of rainstorm kinetic energy. Does the rainfall erosivity index not depend on soil type, land use and orography? R1C2 Response: Thanks for this comment. In the revised manuscript, we modified the Introduction section in order to more clearly describe the rainfall erosivity parameter. Please note that rainfall erosivity represents only the driving force of rainfall on soil erosion and other factors that control soil erosion rates (e.g., soil, land use, topography) are described using other factors and not with rainfall erosivity. These factors are not the scope of this study.

R1C3: - The introduction clearly describes the aim of this manuscript. As far as I am aware, this is the first time that a pan-European radar dataset is used for erosivity assessment. You mention one other study that uses ground-based weather radar data for erosivity assessement. Are there other studies that use radar data for this purpose? If yes, add references. If no, the uniqueness of this manuscript in using radar data may be emphasized more in the introduction.

R1C3 Response: Thank you. We included additional references related to rainfall erosivity estimation using radar-based approaches in the revised version of the manuscript. We also highlighted the uniqueness of this study based on its geographical scope. To our knowledge, while national assessments of rainfall erosivity using ground radar exist, a pan-European attempt which combines data across multiple countries does not.

R1C4: - L. 45/46: This needs rephrasing: "rain gauges are fundamentally represent point scale measurements with a limited cover limits"

R1C4 Response: Thanks for this comment. This sentence was rephrased in the revised version.

R1C5: - L. 70: What is meant by "sub-timestep rainstorm intensity"? Being smaller than the time step of the available precipitation dataset?

R1C5 Response: Thanks for the comment. Here we meant sub-hourly intensity. This was modified in the revised version.

**Data and methods:**

R1C6: - L. 90-91: Are the "GloREDa 1.2 average monthly and annual predictions" based on observations, or are these true predictions, i.e., forecasts? My interpretation is that these are observations and that these observations are employed to compute variables such as rainfall erosivity. Perhaps another term than "precitions" is more appropriate then.

R1C6 Response: Thanks for this useful comment. GloREDa gauge sites can be regarded as ground-truth values of rainfall erosivity since they were computed based on the high-frequency rainfall data for multiple locations based on the measured precipitation with gauges. However, the GloREDa database also includes spatially interpolated maps (e.g., annual average, monthly average) that are based on gauge data. Hence, we more clearly describe the different datasets used, their spatial and temporal properties, and to what extent they are based on measurements or predictions (Section 2.1). Additionally, in the revised version of the manuscript we are now more consistent about the terminology used (e.g., predictions, measured values at rain gauges, etc.). Hence, GloREDa can be regarded as true rainfall erosivity values while other methods can be regarded as predictions.

R1C7: - EURADCLIM dataset: Pitfall 1) is actually caused by factors of which "radar beam attenuation, changes in the reflectivity profiles with distance from the antenna" are important ones. These should be mentioned here and not under 3), because these do not generate noise.

R1C7 Response: Thanks for pointing to these EURADCLIM characteristics. Hence, in the revised version the description of pitfalls was modified, pitfalls No. 1 and No. 3 were merged.

R1C8: - EURADCLIM dataset: In my view, pitfall 3) is already covered by pitfall 2), because non-meteorological echoes and other artifacts can be considered noise. Such artifacts may also be due to "hardware related issues, such as calibration errors". So, I suggest to remove pitfall 3), where a phrase such as "hardware related issues, such as calibration errors" could be added to pitfall 2).

R1C8 Response: Thanks, as suggested we removed pitfall No. 3 and rephrased pitfall No. 2.

R1C9: - L. 114: One has to realize that only a limited number of these rain gauges are actually used in the merged product. Partly because not all gauges are available over the entire period, but especially because of the applied 0.25 mm thresholding on 1-h gauge

accumulations.

R1C9 Response: Thanks for this comment; this was included in the description of the EURADCLIM dataset in section 2.2 of the revised version.

R1C10: - L. 133-144: It seems that the 1-h radar accumulations are disaggregated to 30-min accumulations, where different disaggregation schemes were tested. This should be clarified in the text.

R1C10 Response: Thanks for this comment. We clearly described the disaggregation approach and schemes tested. Several changes were made in Section 2.3.

R1C11: - Not begin familiar with rainfall erosivity computations: would it be possible to add an appendix with the most important equations and description of the computations?
R1C11 Response: In the revised version, we included the basis equations of the rainfall erosivity calculations that we used (in the supplement).

R1C12: - Would it be possible to derive EI60 instead of EI30 based on the GloREDa dataset and EURADCLIM dataset? This would avoid the disaggregation of radar data to 30 min, and hence avoid introducing uncertainty on the intensity over 30-min intervals. I guess that the 30-min interval is relevant because higher rainfall intensities causing stronger erosion are better captured, whereas these may be averaged out too much at the 60-min interval.

R1C12 Response: In the scope of the (R)USLE soil erosion models, the rainfall erosivity (R-factor) is calculated as the product of maximum 30-min rainfall intensity (I30) and rainfall kinetic energy (E). Indeed, 30-minutes time step was chosen based on the plot studies in the United States used for the formulation of the model, wherein the EI30 was the best performing statistical index to predict soil erosion. In theory, variants of the EI30 (e.g., EI15 or EI60) may have varying relevance in different climatic regions and different seasons, but to maintain a consistent comparison between GloREDa (EI30-based) and EURADCLIM, we consistently apply the EI30. In case that only hourly data is available, one can attempt to correct the final rainfall erosivity using empirically derived conversion factors as described in section 2.3. However, in our case study this approach yielded higher bias in the rainfall erosivity estimation as described in section 2.3. In the revised version, this is more clearly described.

**Results and discussion:**

R1C13: - L. 158-159: Note that at least Italy has some coverage, so perhaps write "limited to countries with (almost) full coverage".

R1C13 Response: Thanks, this sentence was rephrased as suggested by the reviewer.

R1C14: - L. 164: "reduced" is used twice in this "reduced percent bias reduced". And

replace "THis" by "This".

R1C14 Response: These suggested corrections were implemented in the revised version.

R1C15: - L. 163-165: Why would the different radar antennas and the lower rain gauge network density lead to overestimation of erosivity? This could be discussed in more detail, for instance, by referring to results shown for EURADCLIM version 1 (Overeem et al., 2023). There, Fig. 7h show large underestimations and overestimations for regions with low gauge network density (leave-one-out-statistics), which could be a proxy for underestimations in specific areas far away from gauges (note that this underestimation largely disappears for the dependent verification in Fig. 7i). Of course, this concerns all 1-h accumulations and not only the extreme ones that are most relevant for erosion. R1C15 Response: Thanks for this comment. This sentence was removed from the revised version of the manuscript.

R1C16: - "artifact" and "artefact" are used interchangeable, choose one of them. R1C16 Response: Thanks, this was corrected in the revised version.

R1C17: - Figure 4: Given the fact that underestimations are found in EURADCLIM version 1 (Overeem et al., 2023; and this will also be the case in version 2) for some regions, such as parts of Norway, Sweden and Austria, it is apparent that few areas with too low erosivity are found when compared to erosivity computed with GloREDa. Of course, these are based on all 1-h accumulations, and the focus in Overeem et al. (2023) is less on 1-h extremes. In addition, extreme rainfall tends to be stronger at the point scale compared to the radar pixel scale (think of areal reduction factors), implying that radar data could underestimate erosivity for the highest rainfall intensities for short duration rainfall, such as 30 min (this radar precipitation estimation could be in the order of 10%). Can you comment on this? R1C17 Response: Thanks for this comment. Figure (4) aims to show the difference in the R-factor when using I30 threshold values of 80 mm/h and 20mm/h in the calculation of EI30, to limit the propagation of error into the quantification of El30. So, the intention is to show the impact of modifications to the method, without giving a comparison against GloREDa in this case. We agree that disparities between the point and grid scale are a fundamental consideration for rainfall erosivity and areal smoothing could naturally cause a numerical bias when comparing gauge and RADAR based acquisitions. However, in this figure we show that the disparities between the EURADCLIM and GloREDa R-factor are dominated by improbably high I30 values when using EURADCLIM in our method configuration. However, please note that this aspect is discussed in relation to Figure 1 where changes were made and additional discussion was added to the manuscript.

R1C18: - Figure 5: Suggest to make this a square plot, because axes have the same scale. This would increase readability. In addition, this could be made more quantitative by incorporating a metric such as the Pearson correlation coefficient.

R1C18 Response: Thanks for this suggestion. Figure 5 was modified as suggested, and R2 values were added to the plots.

R1C19: - Although it is to be expected and confirmed that remaining non-meteorological echoes in EURADCLIM can give rise to erosivity overestimation in several regions, being consistent with Figure 9g,h in Overeem et al. (2023) for EURADCLIM version 1, the red areas with overestimation in Figure 4 are quite abundant. Could this also be related to underestimation by the gridded GloREDa dataset? Rain gauges only sample a limited number of locations. Hence, much more extreme events may occur between rain gauge locations, that may be captured by radars though.

R1C19 Response: Thanks for this comment, we agree with the points raised by the Reviewer #1. By including a data source ensemble (Figure 6) we aim to communicate how multi-modal, multi-method retrievals of rainfall erosivity, can eventually give more robust insights for soil erosion risk assessment. Capturing localized extreme events which have lasting impacts on the long-term annual average rainfall erosivity is particularly relevant. Due to the importance of convective rainfall, these patterns may be highly localized which justifies some of the spatial patterns of standard deviation introduced by EURADCLIM (Figure 6). In the revised version we enhanced the discussion about EURADCLIM-GloREDa, contextualized them, and also mentioned potential GloREDa limitations (e.g., missing extreme events due to limited rain gauge coverage, etc.).

R1C20: - Caption Figure 6: Since four and five datasets are mentioned, respectively, it seems a 4 and 5 "data source ensemble".

R1C20 Response: Thanks, this error was corrected in the revised version.

R1C21: - Figure 6: Why the mean annual erosivity becomes generally lower in case the EURADCLIM dataset is added? Given the overestimation found for EURADCLIM, I would expect higher values? Or does this imply that the EURADCLIM dataset has relatively modest erosivity compared to the satellite-based datasets?

R1C21 Response: Thanks for pointing this out. The patterns are indeed due to EURADCLIM producing relatively modest values for the R-factor in most areas but inducing a high level of spatial variability into the R-factor. One thing to mention is the use of the R-factor calculation using the I30 limit of 80 mm/h for EURADCLIM in the data source ensemble, which causes a reduction in the R-factor. We corrected the explanation in the figure caption and also more clearly explained this phenomenon in the text. Additionally, Figure 6 was also modified. Moreover, changes were also made to Figure 1 and the corresponding discussion.

**Conclusions:**

R1C22: - Why would overprediction occur in European regions with lower radar antenna coverage? A longer distance to a radar would make precipitation estimates less reliable and

can especially make it harder to correct for range dependent sources of error, such as the vertical profile of reflectivity and rain-induce attenuation along the radar beam.

R1C22 Response: Thanks. The conclusions were modified to account for the comment provided by Reviewer #1.

**Figure S3:**

R1C23: - Note that although Austria is covered by weather radars, Austrian radars did not contribute yet to the OPERA data used in EURADCLIM.

R1C23 Response: Thanks for this comment. These characteristics were mentioned in the Figure S3 caption in the revised version.

**Reference list:**

Overeem, A., van den Besselaar, E., van der Schrier, G., Meirink, J. F., van der Plas, E., and Leijnse, H.: EURADCLIM: the European climatological high-resolution gauge-adjusted radar precipitation dataset, Earth Syst. Sci. Data, 15, 1441–1464, https://doi.org/10.5194/essd-15-1441-2023, 2023.

**Anonymous Referee #2, 25 Jun 2025**

R2C1: I was very pleased to read the manuscript "Dynamic assessment of rainfall erosivity in Europe: evaluation of EURADCLIM ground-radar data." The author aimed to evaluate the ground radar-based European RADar CLIMatology (EURADCLIM) precipitation grids to quantify rainfall erosivity across European countries. The manuscript was clear, well-structured, and easy to follow. The multi-scale comparison of erosivity values between EURADCLIM and GloREDa demonstrates the potential application of ground-radar data in soil erosion studies at large time and spatial scales.

I have some comments that could improve the interpretability of the methodology and results. The comments are principally focused on providing more interpretable and understandable results to the readers.

R2C1 Response: We would like to thank Reviewer No. 2 for his/her positive evaluation of our study. Please find below the detailed response to the specific comments provided.

**Abstract**

R2C2: Line 18: Please define KGE because it is the first time that it has been named. R2C2 Response: KGE acronym was defined in the revised version.

R2C3: Lines 26-27. This statement is not explicitly discussed in the manuscript. R2C3 Response: Thanks for this comment, this sentence was removed from the manuscript.

**Introduction**

R2C4: Line 45: Please rephrase the sentence: "... with a limited cover limits predictions..." R2C4 Response: This sentence was rephrased as suggested by the Reviewer No. 2.

R2C5: Line 78, 111, and 348. Please be consistent with the terms. In the document, it is read as "artefacts"

R2C5 Response: The terminology related to terms "artefact" and "artifact" was corrected in the revised version (term artefacts is used in the revised version).

**Methodology**

R2C6: Line 105: Please check the spelling of "kilometres"

R2C6 Response: Thanks, term km is used in the revised version.

R2C7: Line 135 and 136: "Secondly, the measured 30-min rainfall data was aggregated to the hourly time step matching the EURADCLIM resolution, and four rule-based rainfall disaggregation schemes were tested:" This sentence is contradictory; it is not clear if

GloREDa time series were aggregated to hourly time steps or EURADCLIM time series were disaggregated. I think it's the second option, but the sentence is unclear.

R2C7 Response: Thanks for pointing to this unclear sentence, in the revised version we more clearly described the procedure used to disaggregate the data. Hence, firstly the GloREDa data was aggregated from 30-min to 1-h and then different disaggregation schemes were tested to find the most optimal option.

R2C8: Line 99-100: The annual average erosivity map was produced in Panagos et al. (2017) 10.1038/s41598-017-04282-8. In the public repository of GloREDa (Panagos et al., 2013), maps are available only for monthly erosivity. Please clarify which is the source of the annual average erosivity.

R2C8 Response: In the revised version, we more clearly stated that monthly and annual rainfall erosivity maps from Panagos et al. (2023) were used. Several changes were made in section 2.1 to better describe the datasets used.

R2C9: It would be helpful to include a figure in the supplementary material showing the distribution of GloREDa gauge stations around the European countries included in this study. It will familiarize non-European readers with the dataset used in this study. R2C9 Response: In the revised version we included the reference to the paper (Panagos et al., 2015, 2023) where GloREDa stations can be seen. We did not include additional figure with the GloREDa stations since there are already quite a lot figures (in the Supplement as well) and Panagos (2015, 2023) papers are Open access so everyone can check the locations.

**Results and discussion**

R2C10: Figure S2: The authors have related this figure to spatial analysis, for example, in line 16. It will be helpful if at least for the worst predictions (Croatia, Bosnia 160, Herzegovina, Serbia, and Estonia), the points in the figure would have the name of the country. Additionally, the reference to Figure S2 in line 163 suggests spatial variability that is not depicted in Figure S2.

R2C10 Response: Thanks for this comment, countries that are mentioned in the main text are labeled on Figure S2 in the revised version. Figure S2 was modified.

R2C11: Line 164. Please rewrite the sentence "reduced percent bias reduced." R2C11 Response: This sentence was rephrased in the revised version.

R2C12: Line 164-168. It would be helpful for readers to add a map showing the spatial distribution of the OPERA antenna across Europe. Additionally, the map could include a DEM in the background to show the complexity of the topographic conditions in certain regions; it will be helpful to support the statement of lines 166-168.

R2C12 Response: Thanks for this suggestion, in the revised version we made a reference to the paper that shows the locations of OPERA radars (i.e., <a href="https://www.mdpi.com/2073-4433/10/6/320">https://www.mdpi.com/2073-4433/10/6/320</a>) and added discussion about the spatial coverage of these radars (Section 2.2).

R2C13: Figure 1. It would be helpful for non-European readers to include country names in the figure. Maybe adding all names may result in a crowded map; however, it is necessary to include at least those country names referenced in the text (e.g., Finland, Slovenia, Poland, among others).

R2C13 Response: Thanks for this suggestion, but we think that adding countries' names will make the figure unclear. Also adding just a few countries' names would make the figure confusing. Hence, we think that a reader that is not familiar with Europe's geography can check the countries' names on his/her own.

R2C14: Figure S4: The figure's caption is wrong. I guess that GloREDa prediction is on the upper panel, and EURADCLIM is on the lower panel.

R2C14 Response: Thanks for pointing to this error. Figure caption was corrected.

R2C15: Line 213-216. The sentence refers to the spatial distribution of seasonal erosivity; however, Figure S6 does not display spatial characteristics.

R2C15 Response: Thanks for this comment, Figure S6 shows monthly rainfall erosivity values. We rephrased this sentence to make clear what we were referring to.

R2C16: Lines 245-247. The sentence is confusing. It is clear that 50% of the points with more than 10 comparable events obtained KGE greater than 0.4. However, the second part of the sentence looks like a contradiction of the first statement. Additionally, it would be beneficial to include the definition and equation for calculating the Kling-Gupta Efficiency in the methodology section.

R2C16 Response: Thanks for this comment, these sentences were rephrased and a reference to the original paper showing the KGE equation was added.

R2C17: Figure S8. Points and raster should have the same color scale. It will be helpful to compare.

R2C17 Response: Thanks for this comment, Figure S8 was modified as suggested.

R2C18: Figure S12. There are no units in the graph or the caption.

R2C18 Response: Thanks for pointing to this issue, Figure S12 caption was modified, and units were added.

R2C19: Lines 352 -369. Why did the authors not include the comparison of the mean values of the R-factor (ensemble platform) with the GloREDa, which is used as ground truth in this

study? Beyond showing a more detailed resolution of the R-factor, it is important to display the agreement with the reference database. It will be valuable to include this comparison. R2C19 Response: Thanks for this comment, a new figure was created in the revised version according to the suggestions provided by the Reviewer # 2.